# Sensor-Based Gait Retraining Lowers Knee Adduction Moment and Improves Symptoms in Patients with Knee Osteoarthritis: A Randomized Controlled Trial

**DOI:** 10.3390/s21165596

**Published:** 2021-08-18

**Authors:** Sizhong Wang, Peter P. K. Chan, Ben M. F. Lam, Zoe Y. S. Chan, Janet H. W. Zhang, Chao Wang, Wing Kai Lam, Kevin Ki Wai Ho, Rosa H. M. Chan, Roy T. H. Cheung

**Affiliations:** 1Gait & Motion Analysis Laboratory, Department of Rehabilitation Sciences, The Hong Kong Polytechnic University, Hong Kong; tim.wang@postgrad.otago.ac.nz (S.W.); 1155150491@link.cuhk.edu.hk (P.P.K.C.); manfung.lmf@gmail.com (B.M.F.L.); zoe.chan1@ucalgary.ca (Z.Y.S.C.); hanwen.zhang@coloradu.edu (J.H.W.Z.); cwang224-c@my.cityu.edu.hk (C.W.); 2Centre for Health, Activity and Rehabilitation Research, School of Physiotherapy, University of Otago, Dunedin 9016, New Zealand; 3Faculty of Kinesiology, University of Calgary, Calgary, AB T2N 1N4, Canada; 4Department of Integrative Physiology, University of Colorado Boulder, Boulder, CO 80309, USA; 5Department of Electronic Engineering, City University of Hong Kong, Hong Kong; rosachan@cityu.edu.hk; 6Li Ning Sports Research Centre, Li Ning (China) Sports Goods Co., Ltd., Beijing 101111, China; gilbertlam@li-ning.com.cn; 7Department of Orthopaedics & Traumatology, The Chinese University of Hong Kong, Hong Kong; kevinho@cuhk.edu.hk; 8School of Health Sciences, Western Sydney University, Penrith, NSW 2751, Australia

**Keywords:** knee osteoarthritis, medial knee loading, gait retraining, machine learning

## Abstract

The present study compared the effect between walking exercise and a newly developed sensor-based gait retraining on the peaks of knee adduction moment (KAM), knee adduction angular impulse (KAAI), knee flexion moment (KFM) and symptoms and functions in patients with early medial knee osteoarthritis (OA). Eligible participants (*n* = 71) with early medial knee OA (Kellgren-Lawrence grade I or II) were randomized to either walking exercise or gait retraining group. Knee loading-related parameters including KAM, KAAI and KFM were measured before and after 6-week gait retraining. We also examined clinical outcomes including visual analog pain scale (VAS_P_) and Knee Injury and Osteoarthritis Outcome Score (KOOS) at each time point. After gait retraining, KAM_1_ and VAS_P_ were significantly reduced (both *Ps* < 0.001) and KOOS significantly improved (*p* = 0.004) in the gait retraining group, while these parameters remained similar in the walking exercise group (*Ps* ≥ 0.448). However, KAM_2_, KAAI and KFM did not change in both groups across time (*Ps* ≥ 0.120). A six-week sensor-based gait retraining, compared with walking exercise, was an effective intervention to lower medial knee loading, relieve knee pain and improve symptoms for patients with early medial knee OA.

## 1. Introduction

As global aging is poised to a significant healthcare challenge, osteoarthritis (OA) has become a major cause of pain and disability [1]. The knee joint is a vulnerable joint to OA. When compared with the lateral compartment, medial compartment of the knee joint is 4–10 times more likely to be affected by OA, which is mainly caused by the greater knee loading to medial knee [2,3]. The knee adduction moment (KAM) is a commonly accepted surrogate measurement for medial knee loading [4], and the first peak of the KAM (KAM_1_) is positively correlated with the severity [5] and progression [6] of medial knee OA. Comparatively, the clinical significance of the second peak of KAM (KAM_2_) is less well recognized than the KAM_1_ [7].

Given knee loading can be redistributed from medial to lateral compartment of the knee joint, gait retraining has been suggested to lower medial knee loading, and modification of foot progression angle is a commonly adopted strategy [8,9]. It has been reported that individuals with medial knee OA may report an improvement in knee symptoms following a six-week gait retraining to lower medial knee loading by toe-in gait [10]. Furthermore, two recent randomized controlled trials have suggested positive effects of gait retraining [11,12]. Hunt and colleagues found that four-month gait retraining using a uniform toe-out gait modification significantly alleviated knee pain and improved symptoms by reducing medial knee loading [11]. Similarly, Cheung and colleagues showed that individuals with early medial knee OA following a six-week gait retraining displayed reduction in the knee pain and improvement in the physical functions, when compared with participants in the control group who received walking exercise [12]. More importantly, these positive effects can be maintained at least six months after training [12].

However, all these gait retraining programs require sophisticated laboratory equipment, such as motion capturing system and force plates, for evaluation and real-time feedback of walking biomechanics. Given this major limitation, wearable sensors have been developed to measure foot progression angle during walking [13]. However, these devices do not directly measure knee joint loading-related parameters (e.g., KAM). Moreover, the relationship between KAM and foot progression angle is participant-specific and non-linear [14]. Therefore, a personalized gait modification may be more effective than a uniform modification of foot progression angle [14]. Recently, inertial measurement unit (IMU) or wearable sensors have been developed to predict KAM [15,16]. However, these KAM estimations are based on data from healthy participants [15], or did not provide real-time feedback of KAM [16]. Given such limitations, a machine learning algorithm has been recently developed to predict KAM and this KAM estimation algorithm was developed using data mainly collected from patients with knee OA (Kellgren & Lawrence grade I and II) and achieved an excellent agreement (R^2^ = 0.956) with the ground-truth [17]. A real-time feedback interface was also developed for gait retraining [17]. However, its clinical efficacy has yet been examined. Additionally, the effectiveness of gait retraining on patients with early-stage knee OA is insufficient [9].

Hence, this randomized controlled trial compared the efficacy between walking exercise and sensor-based gait retraining using this newly developed KAM prediction algorithm [17], on KAM, knee symptoms and functions in patients with early medial knee OA. We hypothesized that sensor-based gait retraining would lower the KAM_1_ rather than KAM_2_, alleviate symptoms and improve functions, when compared with walking exercise. Considering the knee adduction angular impulse (KAAI), which represents the cumulative KAM over the stance [18], was positively correlated with the cartilage damage [19], and the knee flexion moment (KFM) may also affect medial knee loading after gait retraining [20,21], we also included KAAI and KFM as secondary outcomes. Following gait retraining, we expected that the KAAI would be reduced in gait retraining but not in walking exercise group, and the KFM would not change in both groups as the gait retaining did not target a KFM reduction.

## 2. Patients and Methods

### 2.1. Study Design

This was a prospective, participant-blinded, parallel-group randomized controlled trial comparing walking exercise with sensor-based gait retraining in participants with early medial knee OA. The clinical trial was registered in the HKU Clinical Trials Registry (Reference number HKUCTR-2487). This study was conducted as per the principles of the Declaration of Helsinki and the experimental procedures were reviewed and approved by the Departmental Research Committee, Department of Rehabilitation Sciences, the Hong Kong Polytechnic University.

### 2.2. Sample Size Estimation

The sample size was calculated using G*Power (Version 3.1, University of Kiel, Germany) using data from a previous lab-based gait-retraining study assessing the effect of a six-week gait retraining program on reducing KAM in patients with medial knee OA and reported an effect size of 1.01 [12]. Given the potential discrepancy in KAM measurement between traditional motion capturing and wearable sensor, we opted for a conservative approach and expected a smaller effect size of 0.80 for this sensor-based gait retraining (which is 20% less than the effect size reported by Cheung et al. [12]). Assuming alpha of 0.05 and power of 0.8, 26 participants per group was adequate to power this study. In order to account for a 20% dropout rate in a longitudinal study, 32 participants per group, for a total of 64 participants, were required.

### 2.3. Participants

Participants were recruited from out-patient clinics of the Department of Orthopaedics & Traumatology by a rheumatologist from January to August 2019. Inclusion criteria were: (1) participants with early medial knee OA confirmed by an X-ray (i.e., Kellgren & Lawrence grade I or II); (2) participants with self-reported knee pain at least one day per week during each of the eight weeks prior to participation [10]; (3) participants aged between 40–70 years and they were able to walk unaided for at least one hour. Participants were excluded if they had: (1) body mass index > 35; (2) known learning disability; (3) used shoe insert or knee brace; and (4) corticosteroid injection or serotonin treatment within the previous eight weeks. Patients were screened by an orthopaedic surgeon from a local hospital. All participants who met the inclusion criteria and provided written informed consent underwent a baseline measurement, including both biomechanical and clinical assessments.

### 2.4. Baseline Measurement

The targeted knee was selected for biomechanical and clinical assessments and further analysis. The targeted knee was defined as the affected knee for participants with unilateral knee OA or the most painful side for participants with bilateral knee OA. To minimize the influence of footwear, participants were tested in standardized shoes (Figure 1) during data collection and training. A side pocket at the lateral ankle of each test shoe contained the inertial measurement unit (IMU) sensor (DA14583, Dialog Semiconductor, Reading, UK), which comprised a gyroscope (range: ±2000°/s), an accelerometer (range: ±4 g), operating at 100 Hz.

#### 2.4.1. Biomechanical Assessment

Reflective markers were placed at specific body landmarks according to the calibrated anatomical system technic (CAST) marker set [22,23]. Kinematic data were recorded with a 10-camera motion capture system (Vicon, Oxford Metrics Group, Oxford, UK) sampling at 100 Hz and kinetic data were collected using a force plate (FP4060, Bertec Corp., Columbus, OH, USA) sampling at 1000 Hz. All the participants were asked to walk naturally at their preferred speed with the sensor embedded test shoes [17]. We measured natural walking speed using photo gates (Smartspeed, Fusion Sport, Broomfield, CO, USA) from five successful trials before data collection. We discarded data if the walking trial speed exceeded 5% difference from participant’s mean pace. We then used the average speed for treadmill training (either gait retraining or walking exercise) and reassessment, as KAM can be influenced by walking speed [24].

The kinematic and kinetic data were filtered using a fourth-order, phase-corrected, Butterworth, lowpass filter at 8 Hz and 50 Hz, respectively [12]. The kinematic and kinetic data were processed using Visual3D (C-Motion, Germantown, MD, USA). Joint angle data was derived from the motion capture system using an X-Y-Z rotation sequence. Joint moments were expressed as external moments, referenced about the proximal end of the distal segment. The KAM_1_ and KAM_2_ were extracted from the KAM peak during 0–50% and 51–100% of stance phase, respectively [18]. The peak of KFM was extracted from the KFM peak during stance phase [25]. The KAAI was the integration of the KAM during stance phase [18]. The foot progression angle was defined as the angle between the axis of foot and the direction of progression [26]. Toe-in and toe-out gaits were respectively defined as a decrease and increase in foot progression angle during natural walking [25]. KAM_1_, KAM_2_, peak of KFM and KAAI during stance were averaged and normalized with body mass and height from the five trials.

#### 2.4.2. Clinical Assessment

We measured symptoms and functions using the Chinese version of Knee Injury and Osteoarthritis Outcome Score (KOOS_5_), including five subscales [27], i.e., pain, symptoms, activities of daily living (ADL), sports and recreational function (Sport/Rec) and knee-related quality of life (QOL). Standard answer options were given and each question score ranged from 0 to 4. A normalized score was calculated for each subscale, while 0 indicated the worst symptom and 100 indicated symptom-free or the best condition [28]. We also used visual analog pain scale (VAS_P_) to measure immediate response after the walking test. The VAS_P_ ranged from 0 (no pain) to 10 (the worst pain imaginable).

### 2.5. Randomization

After the baseline measurement, all participants were randomly assigned to either walking exercise group or gait retraining group in a 1:1 ratio. The randomization schedule was computer-generated by a research assistant and the participants were given these randomly generated allocations within sealed opaque envelopes, following the baseline assessment.

#### 2.5.1. Gait Retraining Group

Participants in the gait retraining group received a six-week gait retraining program for KAM modulation according to a previously established protocol [10,12]. In brief, the participants received six sessions of gait modification over six weeks (one session per week). The real-time feedback of KAM was calculated by Visual3D software using force plate and VICON data in the previous study [12], while the KAM feedback used in the present study was estimated by IMU sensors using artificial neural network (ANN), which has been widely used in healthcare applications with excellent performance [17]. During the training, the participants walked on a regular self-paced treadmill (t10.8, FreeMotion, Logan, UT, USA). The KAM curve of the targeted knee was displayed in real time using a tablet device (Figure 2a). The participants were asked to adjust the foot progression angle so that the KAM peak could be reduced by 20%, from the baseline value and the target was indicated by a line on the KAM-time graph (Figure 2b,c). The training time was incrementally increased by three minutes in each session, i.e., from 15 min (first session) to 30 min (last session) over the six sessions. Similar to the previous study, the feedback was progressively removed in the last four sessions [12]. Additionally, the participants were advised to maintain their new gait pattern during daily living after the training.

#### 2.5.2. Walking Exercise Group 

Participants in the walking exercise group were required to perform self-paced walking exercise on the same treadmill with the same test shoes. The training time per session and total training time were identical to that the gait retraining group. However, no feedback about their walking mechanics was provided.

### 2.6. Reassessment

All the participants were reassessed one week after the training. The reassessment was identical with the baseline measurement. All participants’ natural walking speed was controlled using photo gates (Smartspeed, Fusion Sport, Broomfield, CO, USA) to keep the speed between baseline and reassessment was within 5% difference. In addition, we asked participants if they encountered any adverse events related to the intervention.

### 2.7. Statistics

We tested the data normality by Shapiro-Wilk tests. If the data were normally distributed, independent *t*-tests were used to compare the descriptive characteristics between the two groups. Otherwise, Mann-Whitney tests were used. Chi-square tests were employed to test the between-group differences in gender and Kellgren & Lawrence grade. All outcome analyses followed per protocol approach. A 2 × 2 mixed factor ANOVA was performed to compare the interaction effect of groups (walking exercise vs. gait retraining) and time (baseline assessment vs. reassessment) on KAM, KAAI, KFM, VAS_P_ and KOOS scores. Post-hoc tests with Bonferroni adjustment were performed if indicated. Effect sizes were calculated for ANOVA and *t*-test comparisons using $\eta_{p}^{2}$ and Cohen’s d respectively. $\eta_{p}^{2}$ of 0.01, 0.09, and 0.25 were respectively interpreted as small, medium and large effects [29], while Cohen’s d of 0.2–0.4, 0.4–0.8 and >0.8 were respectively interpreted as small, moderate and large effects [30].

## 3. Results

89 patients with medial knee OA were screened, and 71 were eligible for this study (Figure 3). After randomization, 36 and 35 patients were respectively allocated to the walking exercise and gait retraining groups. 62 of 71 participants completed all follow-up training and reassessments while nine participants dropped out due to personal reasons or scheduling conflicts. The two groups were matched in terms of demographics, baseline biomechanical and clinical parameters (Table 1). However, the stance time in gait retraining group was significantly shorter than that in the walking exercise group (*p* = 0.018).

Participants in both groups reported no adverse training effects. We found a significant group and time interaction effect on the foot progression angle (*F_1,60_* = 14.711, *p* < 0.001, $\eta_{p}^{2}$ = 0.197). The foot progression angle was significantly lower (i.e., more toe-in) in the gait retraining group than walking exercise group after training (mean difference = −5.1° (95% CI: −8.5, −1.7), *p* = 0.004, Cohen’s d = 0.75). Pairwise comparison indicated that foot progression angle was significantly reduced by 6.3° (95% CI: −9.3, −3.4) in gait retraining group after training (*p* < 0.001, Cohen’s d = 0.88), but remained similar in walking exercise group (*p* = 0.335). We found no significant group and time interaction effect on the stance time (*F_1,60_* = 0.592, *p* = 0.445, $\eta_{p}^{2}$= 0.010). After gait retraining, the stance time in the gait retraining group was lower than the walking exercise group (mean difference = 0.047 (95% CI: 0.0064, 0.087) *p* = 0.024, Cohen’s d = 0.58). Pairwise comparisons indicated that stance time did not change before and after training in both gait retraining and walking exercise groups (*p* = 0.775 and 0.459, respectively). Most of the participants in gait retraining group adopted toe-in gait to lower the KAM_1_, while five out of 31 (16.1%) patients with medial knee OA used toe-out gait to reduce the KAM_1_ when completed gait retraining (Appendix A).

### 3.1. Primary Outcomes

We found a significant group × time interaction on KAM_1_ (*F_1,60_* = 22.002, *p* < 0.001, $\eta_{p}^{2}$ = 0.268), KOOS_5_ (*F_1,60_* = 4.779, *p* = 0.033, $\eta_{p}^{2}$ = 0.074) and VAS_P_ (*F_1,60_* = 5.182, *p* = 0.026, $\eta_{p}^{2}$ = 0.079), but not KAM_2_ (*F_1,60_* = 0.199, *p* = 0.657, $\eta_{p}^{2}$ = 0.007). After gait retraining, between-group comparisons indicated that KAM_1_ and VAS_P_ in the gait retraining group were significantly lower than the walking exercise group (mean difference = −0.39 (%BW*ht) (95% CI: −0.66, −0.12) and −1.43 (95% CI: −2.55, −0.30), *p* = 0.005 and 0.014, Cohen’s d = 0.73 and 0.63 respectively, Figure 4 and Figure 5), and KOOS_5_ in the gait retraining group was significantly higher than the walking exercise group (mean difference = 8.72 (95% CI: 2.48, 14.97), *p* = 0.007, Cohen’s d = 0.70), while KAM_2_ remained similar between the two groups (*p* = 0.483). Pairwise comparisons showed that following training, KAM_1_ and VAS_P_ significantly reduced by 16.5% ((95% CI: 11.2%, 21.7%), *p* < 0.001, Cohen’s d = 0.90) and 35.5% ((95% CI: 21.8%, 49.2%), *p* < 0.001; Cohen’s d = 0.73) respectively in the gait retraining group, but remained similar in walking exercise group (*Ps* ≥ 0.488). In addition, KAM_2_ remained similar in both groups (*p* ≥ 0.499).

### 3.2. Secondary Outcomes

We found no significant group x time interaction on KAAI (*F_1,60_* = 1.160, *p* = 0.286, $\eta_{p}^{2}$ = 0.019) and KFM (*F_1,60_* = 0.391, *p* = 0.534, $\eta_{p}^{2}$ = 0.006). Following training, KAAI in the gait retraining group was lower than the walking exercise group (mean difference = 0.20 (%BW*ht*s) (95% CI: 0.05, 0.36) *p* = 0.012, Cohen’s d = 0.64, Figure 4), while no significant differences were found on the KFM between the two groups (*p* = 0.447). Pairwise comparisons indicated that KAAI and KFM did not change before and after training in both gait retraining and walking exercise groups (*p =* 0.120 and 0.748, respectively).

Regarding KOOS sub-scores, we found a significant group x time interaction on pain (*F*_1,60_ = 8.540, *p* = 0.005, $\eta_{p}^{2}$ = 0.125), symptoms (*F*_1,60_ = 4.250, *p* = 0.044, $\eta_{p}^{2}$ = 0.066) and ADL sub-scores (*F*_1,60_ = 4.160, *p* = 0.046, $\eta_{p}^{2}$ = 0.065). However, there was no interaction effect on Sport/Rec (*F*_1,60_ = 2.949, *p* = 0.091, $\eta_{p}^{2}$ = 0.047) and QOL sub-scores (*F*_1,60_ = 0.213, *p* = 0.646, $\eta_{p}^{2}$ = 0.004). KOOS pain, symptoms, ADL and QOL sub-scores in the gait retraining group were significantly greater than the walking exercise group after training (mean difference = 7.57 (95% CI:1.36, 13.78), 7.94 (95% CI: 0.11, 15.76), 7.45 (95% CI: 1.48, 13.43), and 9.57 (95% CI: 1.33, 17.82), *p* = 0.018, 0.047, 0.016 and 0.024, Cohen’s d = 0.61, 0.51, 0.63, and 0.59 respectively, Figure 5). Pairwise comparisons indicated that pain, symptoms, ADL and Sport/Rec sub-scores were significantly improved in gait retraining group (mean difference = 6.39 (95% CI: 3.16, 9.62), 5.98 (95% CI: 1.70, 10.27), 3.42 (95% CI: 1.10, 5.74), and 4.46 (95% CI: 1.49, 9.43), *p* < 0.001, =0.008, 0.005, and 0.030, Cohen’s d = 0.61, 0.38, 0.41 and 0.37, respectively), but not in the walking exercise group (*Ps* > 0.721).

## 4. Discussion

The overall aim of this study was to compare the efficacy of a novel sensor-based gait retraining with walking exercise in patients with early medial knee OA. In support of our original hypotheses, we found that real-time feedback gait retraining was an effective and safe treatment for lowering KAM_1_, improving knee pain and enhancing functions in patients with early medial knee OA. In view of the fact that the duration of the intervention of this study was only six weeks and there was no further follow-up, the application of this sensor-based gait retraining should be used with caution.

Foot progression angle was reduced by 6.3° in gait retraining group following training. The result was aligned with the findings from Shull et al. that at least five degrees reduction in the foot progression angle was required to lower KAM_1_ [25]. Most previous gait retraining studies were laboratory-based and the participants received standard instructions to modify foot progression angle [11,25]. For instance, Shull et al. [25] used the toe-in gait and Hunt et al. [11] suggested toe-out gait for patients with medial knee OA to improve knee pain by lowering medial knee loading. However, patients with medial knee OA in gait retraining group in our study were only instructed to modify their foot progression angle according to the visual feedback from the KAM curve, and the foot progression angle modification was individualized.

In the present study, the real-time feedback gait retraining significantly reduced the KAM_1_ by 14.6% after a six-week gait retraining. This finding was in accordance with previous laboratory-based gait retraining studies [10,12]. However, the reduction of KAM_1_ reported by those studies was approximately 20–22% [10,12], which appeared to be more effective than the present method. This discrepancy may be explained by the single strategy (i.e., adjustment of foot progression angle) employed in the current study. Although most participants in the previous studies also modified foot progression angle [10,12], they were allowed to adopt other gait modifications, such as trunk sway and knee thrust gait, which can lower KAM_1_ [31]. A second reason for a lower KAM_1_ reduction could be due to the device accuracy in predicting KAM [17]. Even most participants were able to reduce >20% KAM during the visual feedback gait retraining, such performance only translated to an actual 14.6% reduction, which was below the target and could be related to measurement error [17]. The 14.6% reduction of KAM_1_ and improvements of knee symptoms are of clinical relevance. Aligned with our hypothesis, the KAM_2_ remained similar in both groups after training, and this finding was in consistent with previous studies [20,25]. This could be explained by the fact that most of the participants (83.9%) in the gait retraining group adopted a toe-in gait, which has been shown not to change KAM_2_ [25].

After gait retraining, KAAI was significantly reduced when compared with walking exercise group. This finding needs to be interpreted with caution, as the stance time in gait retraining group was significantly shorter than in walking exercise group, despite the controlled walking speed in both baseline assessment and reassessment. However, KAAI did not change in walking exercise and gait retraining groups across time. The unchanged KAAI in walking exercise group could be explained by no gait information being given during walking. The unchanged KAAI in the gait retraining group could be explained the target parameter of the gait retraining program. Participants focused on the KAM peak instead of the areas under the curve. In addition, the walking speed and stance time were similar between baseline assessment and reassessment, thus the KAM magnitude at mid-stance (the valley between KAM_1_ and KAM_2_) may be increased, and create a less distinct double-peak waveform.

Similar to previous laboratory-based gait retraining studies [10,12], the sensor-based gait retraining in our study did not reduce KFM after gait retraining. This could be explained by the absence of KFM information in our gait retraining protocol. It indicated that the real-time feedback gait retraining in our study is an effective intervention to lower medial knee loading, though a previous study reported that the KAM_1_ reduction might not guarantee a reduction of medial knee loading, as the peak KFM increased during gait retraining [21].

Consistent with previous studies, patients with medial knee OA in gait retraining group reported that the VAS_P_ and KOOS_5_ score were significantly improved [12,32]. However, the improvements of VAS_P_ and KOOS_5_ are only 1.5 and 4.5, which are less than the minimal clinically important difference (MCID) (i.e., 1.8 and 10 respectively) for patients with knee OA [33,34]. This may be explained by the potential floor effect, as we did not set a minimum pain score of VAS_P_ and KOOS_5_ in the inclusion criteria. Although walking exercise was reported to be beneficial for patients with knee OA [35,36], we did not observe any improvement of pain and functions in the walking exercise group. Patients in the walking exercise group did not receive any instructions to lower KAM during walking. Gait retraining was targeting motor learning rather than providing physical training. Not like previous studies [35,36], both participants in gait retraining and walking exercise groups were not asked to walk more outside the lab in their daily life. Therefore, patients in the walking exercise group may not demonstrate any significant improvement.

In partial support of our hypothesis, most of the KOOS subscales, including pain, symptoms, activities of daily living and quality of life were significantly improved in gait retraining group. Presumably, the reduction of KAM_1_ unloaded the medial knee loading, and consequently relieved pain and symptoms, as well as improved knee-related quality of life. However, we did not identify any change in the KOOS function in sport and recreation. Additionally, the KOOS knee-related quality of life was similar in gait retraining group after gait retraining. This finding might be explained by the gait retraining used in our study mainly targeted lowering KAM during walking, rather than other sport or recreation activities.

The sensor-based gait retraining allows more practical gait retraining without using any expensive and sophisticated equipment in the laboratory. Furthermore, compared with previous sensor-based gait retraining studies [13,15,16], which only monitored foot progression angle but not KAM, the sensor-based gait retraining used in our study could predict and present real-time feedback of KAM for participants with medial knee OA. This is a viable solution for easier self-management for patients with medial knee OA outside a laboratory environment. However, several limitations should be considered in this study. Firstly, as we only recruited patients with early medial knee OA (Kellgren & Lawrence grade I and II) in this study, the effects of our sensor-based gait retraining for patients with Kellgren & Lawrence grade III and IV knee OA remain unknown. Secondly, participants in gait retraining group were only instructed to modify their foot progression angle. Other modifications, such as lateral trunk lean and knee thrust gait, were not included. Thirdly, the present study only examined the short term (i.e., six-week) effect of the training. The longer-term effect was not investigated. Fourthly, we did not measure physical activity in our participants during the study. Given the fact that the KOOS (ADL) subscale scores were comparable between gait retraining and walking exercise group, we believed the influence of physical activity on the overall findings should not be a concern. Finally, the gait retraining was conducted on a treadmill instead of overground walking. Future studies are warranted to test the clinical effects of the sensor-based gait retraining in the wild.

## 5. Conclusions

The sensor-based gait retraining with real-time visual KAM curve feedback provides an immediate effect to lower KAM_1_ and KAAI, alleviate knee symptoms and enhance functions in patients with early medial knee OA. Our results demonstrated a proof-of-concept to advance the current clinical practice to rehabilitate patients with early medial knee OA without using sophisticated laboratory equipment.

## Figures and Tables

**Figure 1 sensors-21-05596-f001:**
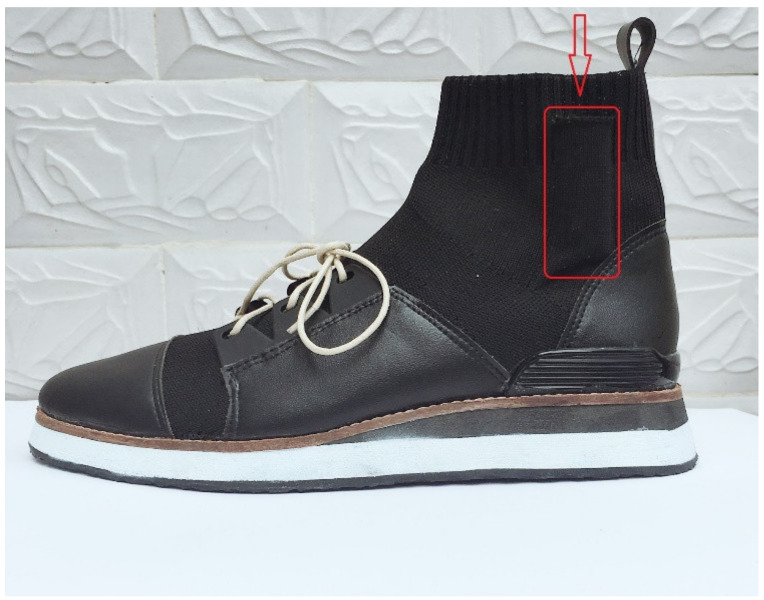
Standardized shoe model used in the present study (the rectangle represents a fit sized pocket to contain the sensor).

**Figure 2 sensors-21-05596-f002:**
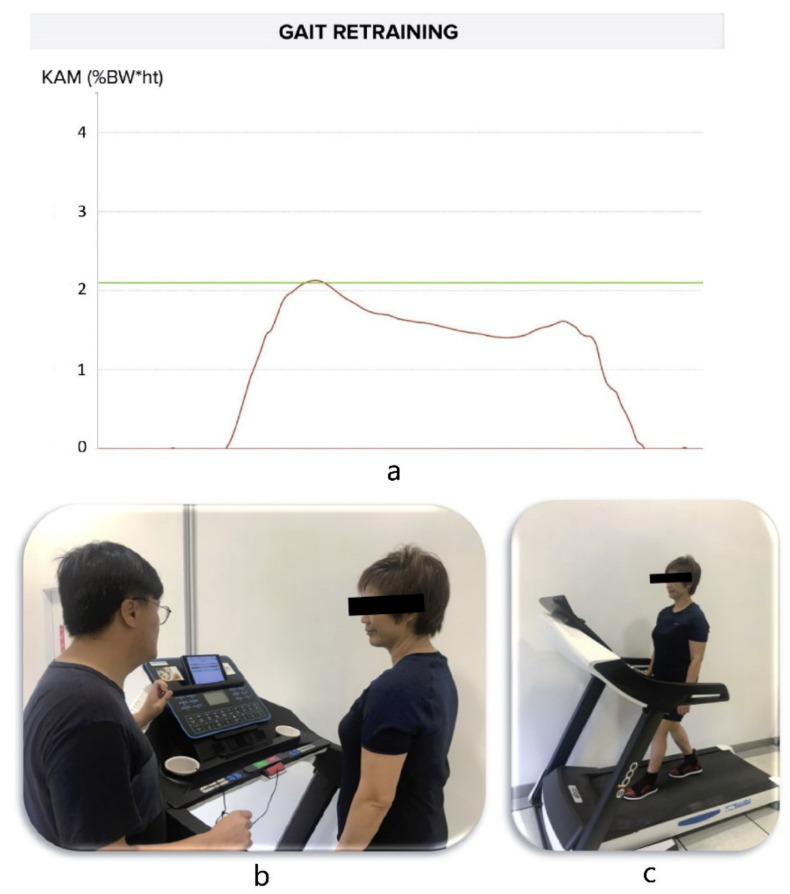
(**a**) Example of the real-time predicted KAM curve (red line) when the participant walked with modified gait on a treadmill. The green line indicates the target threshold of KAM; (**b**) Researcher was explaining the intervention to the participant; and (**c**) Participant was undergoing gait retraining with visual feedback.

**Figure 3 sensors-21-05596-f003:**
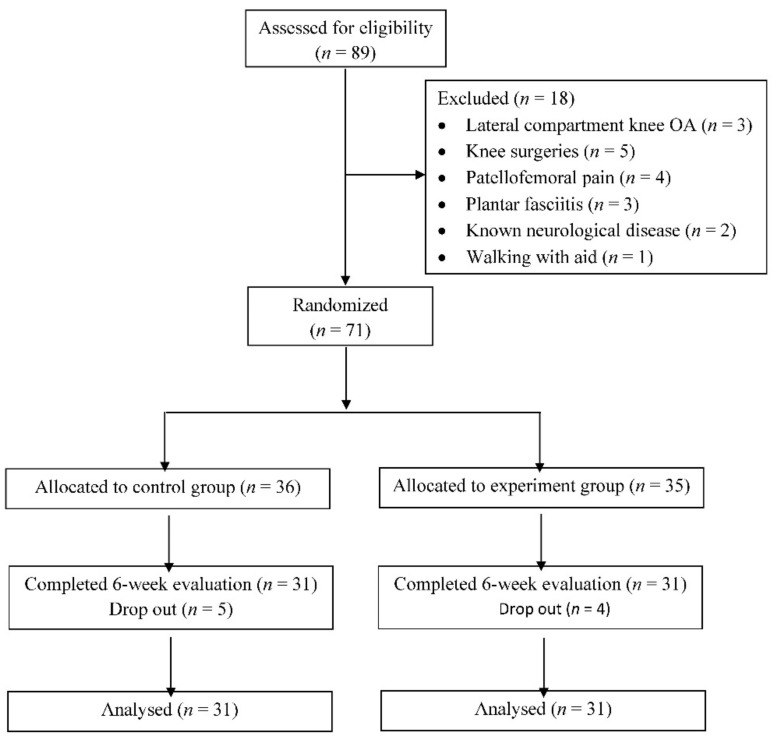
CONSORT flow chart.

**Figure 4 sensors-21-05596-f004:**
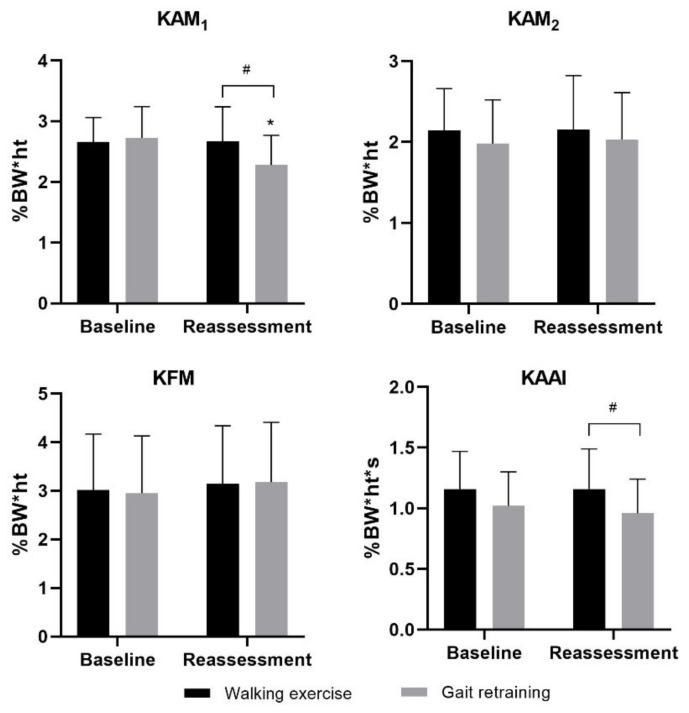
Effects of gait retraining on biomechanical outcomes (KAM, KFM and KAAI). Error bars indicate standard deviations. * indicates significant within-group difference between post- and pre-training score. # indicates significant between-group difference at certain time point.

**Figure 5 sensors-21-05596-f005:**
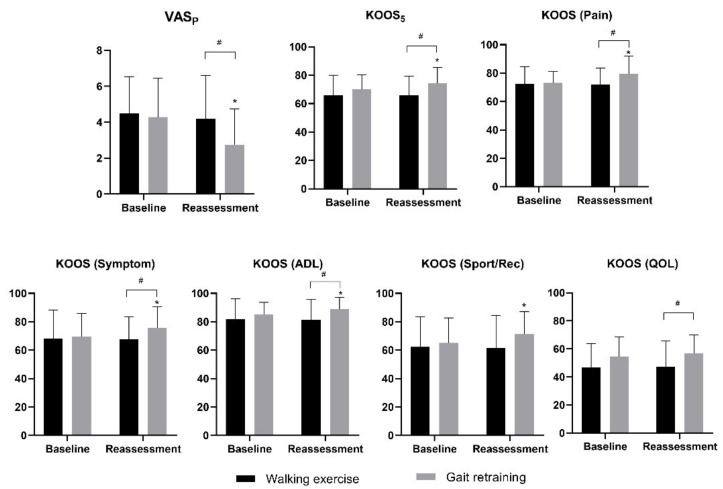
Effects of gait retraining on clinical outcomes (VASP, KOOS_5_, subscales of KOOS scores). Error bars indicate standard deviations. * indicates significant within-group difference between post and pre-training score. # indicates significant between-group difference at certain time point.

**Table 1 sensors-21-05596-t001:** Characteristics of participants.

Group	Gait Retraining Group Mean (SD)	Walking Exercise Group Mean (SD)	*p* Value
Gender (male/female)	15/16	14/17	0.800
Age (year)	59.1 (7.2)	61.7 (6.8)	0.157
Body mass (kg)	62.0 (11.0)	64.2 (10.6)	0.342
Body height (m)	1.65 (0.09)	1.65 (0.08)	0.940
Kellgren & Lawrence grade			1.000
Grade I	15	16	
Grade II	16	15	
Walking speed (m/s)	1.02 (0.15)	1.01 (0.14)	0.793
Stance time (s)	0.711 (0.063)	0.766 (0.108)	0.018
Baseline biomechanics			
FPA (°)	12.1 (6.1)	11.3 (4.4)	0.565
KAM_1_ (%BW*ht)	2.73 (0.51)	2.66 (0.40)	0.548
KAM_2_ (%BW*ht)	1.98 (0.54)	2.14 (0.52)	0.242
KFM (%BW*ht)	3.14 (1.20)	3.02 (1.15)	0.694
KAAI (%BW*ht*s)	1.02 (0.28)	1.16 (0.31)	0.055
Baseline symptoms			
VAS_P_ (0–10)	4.3 (2.2)	4.5 (2.0)	0.683
KOOS			
KOOS_5_ (0–100)	70.0 (10.3)	66.1 (15.0)	0.208
Pain (0–100)	73.1 (7.9)	72.5 (12.1)	0.820
Symptoms (0–100)	69.7 (16.2)	68.1 (20.3)	0.730
ADL (0–100)	85.4 (8.4)	81.9 (14.3)	0.249
Sport/Rec (0–100)	65.2 (17.6)	62.3 (21.2)	0.560
QOL (0–100)	54.6 (14.0)	46.7 (17.2)	0.055

ADL = function in daily living, BW = body weight, FPA = foot progression angle, ht = body height, KAAI = knee adduction angular impulse, KAM = knee adduction moment, KFM = knee flexion moment, KOOS = Knee injury and Osteoarthritis Outcome Score, QOL = knee-related quality of life, Sport/Rec = sports and recreational function.

## Data Availability

The data presented in this study are available on request from the corresponding author. The data are not publicly available due to privacy.

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
