# Peer review of "Sensor-Based Gait Retraining Lowers Knee Adduction Moment and Improves Symptoms in Patients with Knee Osteoarthritis: A Randomized Controlled Trial"

_sensors, 2021, doi:10.3390/s21165596_

Round 1

Reviewer 1 Report

General comments

In the study entitled “Sensor-Based Gait Retraining Lowers Knee Adduction Moment and Improves Symptoms in Patients with Knee Osteoarthritis: A Randomized Controlled Trial”. The study aims to determine the differences on biomechanic performances between walking exercise and sensor-based gait retraining in individuals with early medial knee osteoarthritis. The current study presents potentially useful results that may use in rehabilitation for individuals with KOA. However, kinematics and spatio-temporal variables were not reported in the study. I strongly recommend provide kinematics and spatio-temporal variables to explain the potential mechanism. Certain limitations have also been observed. The specific areas for improvement are described below.

Specific comments

Abstract

Line 28-31: P values should be reported separately; KAM1 and KAM2 need to be distinguished.

Introduction

The introduction about used inertial sensors to improve biomechanics performance in previous studies should be detailed. The rationle in the manuscript was insufficient to support the significance of the newly developed sensor-based gait retraining.

Line 55: “reducing knee loading” should be replaced by “reducing medial knee loading”.

Line 56: “a course of” is not clear, please clarify.

Line 60: “requires” should be replaced by “require”.

Line 67-73: “a machine learning algorithm.[15]”. The findings of this study should be described shortly.

Patients and Methods

Line 91: Please add “sensor-based” before “gait retraining”.

Line 102: Please change “32 participants were required” with “16 participants per group for a total of 32 participants were required”.

Line 118-119: Please change “participant” with “participants”.

Line 120: Standardized shoes was used in current study, what is characteristics of the shoes used?

Line 130:  It is necessary to briefly introduce the type of sensors used in this study, such as the composition of the sensor and location. The validity and reliability of the sensors system should be clarified here.

Line 136: Please explain how to get the joint angle data in detail. Does it derived from sensor system or motion capture system, or both? This sentence is unclear, please clarify

Line 141-142: “KAM1, KAM2, peak of KFM, and KAAI during stance were averaged and normalized with body mass and height from the five trials”. Please add a specific explanation about how these parameters were obtained.

Line 149: “symptoms” should replace by “symptom”; add “the” before “best”.

Line 151: Please add “the” before “worst”.

Line 167: Please explain “incrementally increased”? Pleas clarify the question.

Line 168: In the previous study[12], the feedback was progressively removed in the last three sessions, why did you choose last four sessions?And what is “progressively” specifically means?

Line 178: remove “that of” .

Line 181: Why did the author conduct the reassessment a week later? Did this time strictly controlled for all participants?

Line 190: Please change “group” to “groups”.

Results

Degree of freedom should be reported in F test.

Line 201: Please change “reassessment” with “reassessments”.

Line 231: Please change “ps” with “Ps”.

Line 232: The format of Figure 4 and Figure 5 should be unified, such as the location of the legend.

Line 252: add “and” before “QOL”.

Line 256: add “and” before “Sport/Rec”.

Line 258: add “and” before “4.46”.

Line 260: Please change “P” with “Ps”.

Discussion

Line 266: add “early” before “medial knee OA”.

Line 271-273: Refs should provide to support this claim.

Line 278-280: Does the gait patterns influence KAM? Please explain it or compare it with previous research.

Line 287-290: In the experimental protocol, there were no descriptions or restrictions on other gait modifications. If you want to make a comparison, please mention it here.

Line 291-294: Provide ref. to support this sentence.

Line 295-296: Please give a brief explanation.

Line 298-305: Are the results similar to those of previous studies? “on the other hand...” Please revise the sentence. Why the similar speed would influence the results? Please clarify.

Line 313-315: “In consistent with previous studies, patients with medial knee OA in gait retraining group reported that the VASp and KOOS5 score were significantly improved in the gait retraining group only.” Please delete “in gait retraining group” , references should be provide to support the statement.

Line 316-317: Please change “i.e., 1.8 and 10 respectively” with “(i.e., 1.8 and 10 respectively)”.

Line 323-325: This study didn’t measure the physical activity level of the participants during their daily activities, which may influence the clinical outcomes.

Line 336: I recommended to compare the results of this sensor-based gait retraining study with other sensor-based studies.

Conclusion

Line 353: add “early” before “medial”.

Reviewer 2 Report

General:

This study performed an RCT to compare the effect a gait retraining intervention compared to a walking exercise on the knee adduction moment and clinical scores in patients with knee osteoarthritis. The novelty of the study, the use of sensors to estimate the KAM and provide feedback is insufficiently explained, especially in the methods. Detailed information can be found in the publication of the algorithm, but it’s not entirely clear how it was applied in this study and without this information the study is essentially a copy (with a larger sample size) of a previously published study from the same group (Cheung et al. 2018).

Specific points:

Introduction:

Lines 71ff: It should be mentioned here, that this algorithm (that is probably applied in the current study) is based mainly data from participants with knee OA (K/L I and II). This is a very important information as this would otherwise be a major concern regarding the use of this algorithm in predicting KAM in gait of patients with knee OA

Methods

Lines 98ff: The sample size estimation only partially makes sense. From the presented parameters (10 subjects per group, 20% dropout, total of 32 participant = 16 per group) 16 participants per group should be included. What was the rational of including more than double that number per group for the presented study (36 and 35 patients)? This applies also to the cited paper by Cheung et al. 2018, that presents a very similar study design and sample size calculation but only included 11 subjects per group. Why would you expect a smaller effect in a very similar study design that justifies such a big increase in sample size?

Lines 125 ff: How was anatomical marker placement ensured for the markers on the foot with participants that wore shoes? Differences in marker placement could heavily influence the foot progression angle and is more difficult with shoes because the anatomical landmarks are more difficult to find.

Lines 130ff: How was the self-selected walking speed determined? Was it kept the same from baseline to postintervention? What was the walking speed during the training? What was the walking speed in the postintervention biomechanical measurement?

Lines 159ff: It should be clearly stated how the KAM was estimated and how the present study differs from the study by Cheung et al. 2018 (reference 12). The way the methods are written currently, it is not known how the feedback is generated.

Results

Lines 198ff / Table 1 / Figure 3: In the statistics section (Line 189) you described that your analysis followed an intention to treat approach. However, from the presented results it appears that patients that dropped out during the intervention were excluded from the analysis (which is not according to the intention to treat approach).

Figure 3: The correct term is Plantar fasciitis (not planter fasciitis).

Table 1: Did you measure other spatiotemporal gait parameters such as length/duration of stance phase? This could influence KAAI if one group had a longer or shorter stance phase (despite comparable walking speed).

Lines 214-215: should be changed to: … the foot progression angle was significantly reduced by 6.3°

Lines 221-225: at which time point?

Line 266: “than walking exercise” should be deleted or sentence should be rephrased. There is no comparison in this sentence.

Discussion:

Lines 278-281: This is a result and should be described in the results section and not the discussion. Additionally, it would be interesting to know whether the change in foot progression angle was correlated with the change in other biomechanical parameters such as the KAM.

Lines 288-290: How did you ensure that patients did not adapt other additional strategies (such as trunk sway/knee thrust gait) to lower the KAM during the training session?

Line 299: Do you mean “did not change” (instead of reduce)?

Line 313-314: “in the gait retraining group” is mentioned twice in the sentence.

Round 2

Reviewer 1 Report

Thank you for incorporating reviewers’ comments to improve the manuscript. The authors have adequately addressed my concerns. I am not further comments for the manuscript.